# OSIRIS: Bridging Analog Layout Circuit Design and Machine Learning with Scalable Dataset Generation

**Giuseppe Chiari, Michele Piccoli & Davide Zoni**
DEIB
Politecnico di Milano
Milano, MI 20133, Italy
`{giuseppe.chiari,michele.piccoli,davide.zoni}@polimi.it`

## ABSTRACT

The automation of analog integrated circuit (IC) design remains a longstanding challenge, primarily due to the intricate interdependencies among physical layout, parasitic effects, and circuit-level performance. These interactions impose complex constraints that are difficult to accurately capture and optimize using conventional design methodologies. Although recent advances in machine learning (ML) have shown promise in automating specific stages of the analog design flow, the development of holistic, end-to-end frameworks that integrate these stages and iteratively refine layouts using post-layout, parasitic-aware performance feedback is still in its early stages. Furthermore, progress in this direction is hindered by the limited availability of open, high-quality datasets tailored to the analog domain, restricting both the benchmarking and the generalizability of ML-based techniques. To address these limitations, we present *OSIRIS*, a scalable dataset generation pipeline for analog IC design. *OSIRIS* systematically explores the design space of analog circuits while producing comprehensive performance metrics and metadata, thereby enabling ML-driven research in electronic design automation (EDA). In addition, we release a dataset consisting of 87 100 circuit variations generated with *OSIRIS*, accompanied by a reinforcement learning–driven baseline method that exploits *OSIRIS* for analog design optimization.

## 1 INTRODUCTION

Analog integrated circuits (ICs) are critical in a wide range of modern electronics applications, including telecommunications, power electronics, audio engineering, biomedical engineering, and instrumentation. They enable precise sensing, amplification, and filtering, serving as a critical bridge between digital systems and the physical environment. This capability ensures robust data acquisition and signal processing across various applications. However, unlike digital ICs, which benefit from highly automated and increasingly ML-driven design flows (Razavi, 2005; Carusone et al., 2011; Allen & Holberg, 2011; Cui et al., 2024; Fang et al., 2025; Liu et al., 2025a; Zhao et al., 2025; Liu et al., 2025b), analog circuit design remains largely manual due to *(i)* complexity, unlike digital design based on simple logic gates, analog circuits use diverse components (MOSFETs, resistors, capacitors) with intricate interconnections, *(ii)* sensitivity, small changes can drastically affect functionality, leading to a vast search space. In particular, the analog design flow is divided into front- and back-end phases. The front-end phase targets schematic-level optimization, including component parameter tuning (e.g., transistor sizing) and topology selection. The back-end phase tackles the complex task of translating the optimized schematic into a manufacturable physical layout, encompassing layout generation and feasibility verification. In the front-end phase, designers determine the circuit's functional structure by selecting a topology and sizing devices to meet electrical specifications under idealized or schematic-level models. These tasks operate on abstract representations, focusing on biasing, gain, bandwidth, or noise through symbolic optimization. The back-end phase then converts the validated schematic into a manufacturable physical layout. This requires solving placement and routing under strict geometric constraints, where small

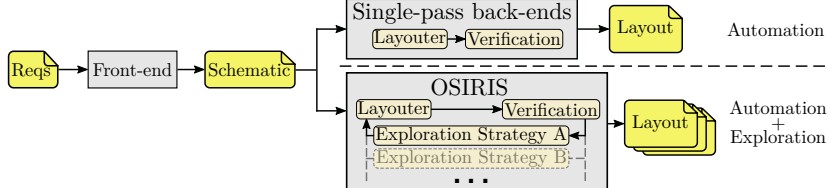

Figure 1: The analog design flow comprises two phases *(i)* front-end, which translates design requirements into a schematic, *(ii)* back-end, which generates the corresponding layout. Unlike existing back-end methods that rely on single-pass automation, *OSIRIS* enables iterative exploration strategies.

spatial modifications can alter matching, introduce parasitics, or break critical symmetries. Designers must manage layout-dependent effects, process variations, and wiring-induced parasitics, ensuring that the final layout passes DRC, matches the schematic through LVS, and undergoes parasitic extraction followed by post-layout validation. Because analog performance is extremely sensitive to layout geometry, this stage is iterative and expert-driven, constituting a major bottleneck. These challenges motivate datasets and frameworks that expose physically accurate, parasitic-aware layout information suitable for ML-based methods. At present, the research effort mainly targets front-end optimizations encompassing the optimization of the topologies of analog circuits leveraging graph neural networks (GNNs), e.g., CktGNN (Dong et al., 2023), large language model (LLM) solutions, e.g., AnalogCoder (Lai et al., 2025a;b), LaMAGIC (Chang et al., 2024), and TopoSizing (Wei et al., 2025) reinforcement learning (RL) e.g., AutoCircuit-RL (Vijayaraghavan et al.) or decoder-only transformers, e.g., EVA (Gao et al., 2025c). Notably, AnalogCoder employs domain-specific prompt engineering and a feedback optimization loop to produce PySpice code, which can be translated into SPICE netlists. LaMAGIC, instead, fine-tunes the Flan-T5 model (Chung et al., 2024) to directly map textual specifications to optimized designs in a single forward pass, mainly targeting power converters. More recently, AnalogGenie (Gao et al., 2025b) introduced a domain-specific GPT-based decoder model that represents each device pin as an individual graph node, enabling fine-grained modeling of circuit connectivity. AnalogFed (Li et al., 2025), an advanced version of AnalogGenie, targets analog topology discovery within a federated learning framework, allowing the integration of multiple private datasets while preserving confidentiality. The inherent complexity (Noori Zadeh & Elamien, 2025) of the analog layout design phase, which is highly sensitive to parasitic effects, tightly bound to specific technology nodes, and constrained by intricate manufacturability requirements, hinders back-end automation. Few seminal works employ GNNs to predict placement performance and guide its optimization (Li et al., 2020) or to increase the accuracy of front-end simulation (Ren et al., 2020). At the same time, variational autoencoders have been applied to analog routing by learning expert-like strategies from manually crafted layouts (Zhu et al., 2019). RL, instead, has been applied to place FinFET modules on a grid, optimizing for symmetry and alignment errors, area, and wirelength (Ahmadi & Zhang, 2021) and to minimize total routing cost in terms of wirelength, number of vias, and design rule violations (Chen et al., 2023a). Despite promising research efforts, back-end automation remains comparatively underexplored. Current analog layout optimization methodologies are human-centric, relying on trial-and-error strategies, making the process time-consuming and resource-intensive. This work introduces *OSIRIS*, a novel end-to-end back-end flow designed to enable scalable machine learning research in analog layout automation. As illustrated in Figure 1, *OSIRIS* differs from existing back-end flows by enabling the instantiation of iterative, performance-driven layout design space exploration strategies rather than single-pass automation. It offers three primary contributions to the state of the art:

- *OSIRIS*: an efficient pipeline to generate, validate, and evaluate large volumes of layouts for generic analog circuits, producing tens of thousands of DRC-clean and LVS-verified designs. A representative dataset, generated by *OSIRIS* comprising 87 100 layouts, is released to demonstrate its capabilities.

- RL-driven optimization: a reinforcement learning framework that leverages *OSIRIS* to iteratively explore the layout design space and optimize circuit implementations based on parasitic-aware performance feedback.

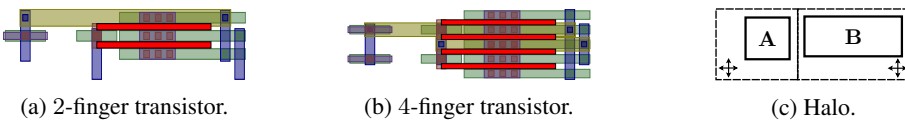

| (a) 2-finger transistor. | (b) 4-finger transistor. | (c) Halo. |

Figure 2: Degrees of freedom explored by *OSIRIS*. (a) a 2-finger transistor, (b) a 4-finger transistor, and (c) a halo wraps each component, allowing it to move freely within the designated space.

- Dataset use case: using the *OSIRIS*-generated dataset to fine-tune an LLM that automatically generates DRC-free and LVS-verified component layouts from sizing specifications.

The *OSIRIS* code and the accompanying dataset are made available online. [1]

## 2 RELATED WORK

The optimization of analog IC physical design is particularly challenging due to tight performance constraints, sensitivity to parasitics, and complex design rules that traditional techniques often fail to handle effectively, requiring time-consuming intervention of expert engineers. Recent advances in machine learning, particularly reinforcement learning and graph neural networks, are being explored to improve placement (Mirhoseini et al., 2021; Li et al., 2020) and routing (Liao et al., 2020b;a), although their application to analog design remains in early development (Huang et al., 2021; Zhu et al., 2019).

**Back-end automation frameworks** The vision of a back-end automated analog IC design flow envisions a seamless pipeline, starting from netlist-level specifications and concluding with a manufacturing-ready physical layout, with each stage executed automatically or with minimal human intervention. This streamlined process seeks to accelerate development and reduce design cycles by integrating diverse tools, established methodologies, comprehensive device libraries, and standardized procedures into a unified framework. The Berkeley Analog Generator (BAG) (Crossley et al., 2013; Chang et al., 2018) enables parameterized generator creation for analog and mixed-signal circuits, automating schematic instantiation and layout synthesis to satisfy specified performance criteria. Despite its utility, BAG depends on predefined templates and often requires substantial domain expertise and parameter tweaking. More recent frameworks adopt a place-and-route-centric paradigm, framing layout generation as an optimization problem. ALIGN (Kunal et al., 2019; Dhar et al., 2020; Sapatnekar, 2023) translates SPICE netlists into GDSII layouts, supporting multiple analog circuit families. Likewise, the open-source MAGICAL framework (Xu et al., 2019; Chen et al., 2020; 2021) offers a fully automated flow from an unannotated netlist to a complete GDSII layout by employing custom place-and-route algorithms and constraint extraction techniques. Despite recent progress, the state of the art lacks back-end automation frameworks specifically designed for iterative design space exploration and layout optimization. MAGICAL has been employed, along with DNN-Opt (Ahmadi & Zhang, 2021) and Bayesian optimization, as the deterministic layouter in closed-loop methodologies that adjust transistor sizing by leveraging post-layout performance information (Budak et al., 2023; Gao et al., 2024). These methodologies are orthogonal to *OSIRIS*, which directly optimizes circuit layouts. Recent efforts have investigated the use of RL with Steiner trees (Basso et al., 2024) and relational GNNs (Basso et al., 2025; Della Rovere et al., 2025) in automatic flows to tackle floorplanning by optimizing wirelength and area consumption. However, the state of the art does not offer frameworks specifically intended to enable adaptive layout refinement through repeated evaluation and learning in parasitic-aware, performance-constrained environments, leaving a critical gap in back-end analog layout design space exploration and optimization. Considering *(i)* the importance and complexity of delivering an optimized backend flow and *(ii)* the lack of layout datasets, *OSIRIS* presents an ML-driven layout generator and dataset than can effectively fuel further research in the design of innovative ML back-end optimization strategies.

**Open analog datasets** Analog circuits datasets are crucial to enable ML-based research in topology generation, specification-driven design, or performance prediction. Recent efforts have begun to curate and label significant front-end analog datasets. The OCB dataset of CtkGNN (Dong et al., 2023) contains 10 000 generated operational amplifiers annotated with device parameters and simulator-

---

[1]Code and dataset: https://huggingface.co/datasets/hardware-fab/osiris

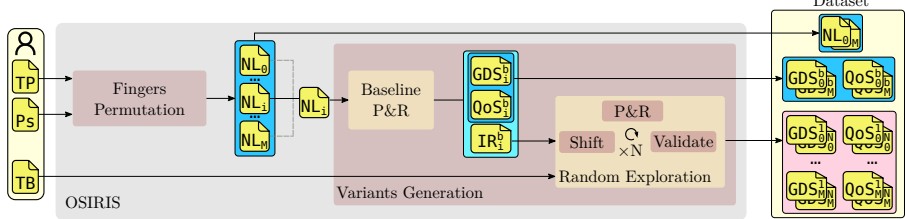

Figure 3: The *OSIRIS* pipeline. It takes as inputs a netlist template (TP), a testbench (TB), and pairs of parameters matching transistors (Ps). For each TP, *OSIRIS* generates M netlists (NL). For each NL, it generates N layout variants (characterized by GDS and QoS files). Therefore, it generates M×N layout variants. It is divided into two stages, *(i) Fingers Permutation* and *(ii) Variants Generation*.

computed gain and bandwidth. ALIGN (Kunal et al., 2019), in addition to a place-and-route framework, offers a diverse set of circuit netlists. More recently, AnalogGenie (Gao et al., 2025a) proposed a dataset comprising 3 350 distinct analog circuit topologies, each labeled with performance metrics, while AMSNet (Tao et al., 2024) provides a set of transistor-level SPICE netlists paired with schematic images. Despite recent progress, the availability of open datasets targeting backend remains highly limited, restricting the practical use of ML techniques. Modern ML methods require large-scale datasets that comprehensively represent the analog circuit design space to learn and identify optimal solutions effectively.

# 3 DATASET GENERATION INFRASTRUCTURE

## 3.1 DESIGN SPACE DIMENSIONS IN DATASET GENERATION

In analog physical layout design, estimating the impact of parasitic elements is critical. These elements arise from layout geometries and the proximity of components, often degrading circuit performance. While increasing the distance between components can reduce parasitic effects, minimizing the total area remains essential for efficient design. To balance these trade-offs, *OSIRIS* is structured around two hierarchical dimensions: *(i)* the number of transistor fingers and *(ii)* component placement. By systematically varying these factors, *OSIRIS* effectively explores the layout design space, capturing the parasitic effects of structural and spatial configurations.

**Number of transistor fingers** In a transistor layout, fingers are the gate segments that share source and drain regions, enabling a single transistor to be split into multiple parallel segments to improve layout compactness, reduce parasitic effects, and enhance performance. Given a circuit netlist, each transistor may support multiple valid finger counts while keeping its width and length fixed. Figure 2a and Figure 2b show examples of a two- and four-finger transistors, respectively; fingers are highlighted in red.

**Component placement** A surrounding boundary, referred to as a *halo*, is placed around each netlist component, i.e., transistors, resistors, and capacitors. This configuration permits unrestricted movement of components within their respective halos. Figure 2c shows the halo mechanism applied to two generic components A and B. Each component is wrapped in a bounding box, which allows it to move freely inside.

*OSIRIS* is designed to be extensible beyond the two degrees of freedom presented here. While the current release explores variability through finger permutations and spatial perturbations, the underlying infrastructure is meant to be easily extended with novel and complex layout transformations

## 3.2 OSIRIS PIPELINE

Figure 3 illustrates the *OSIRIS* dataset generation pipeline. *OSIRIS* aims to systematically generate a diverse and structured collection of layout variants suitable for statistical design analysis, training machine learning models, or exploring layout optimization strategies. The pipeline takes as inputs a circuit netlist template (TP), which specifies the connectivity and device-level specifics (e.g., sizing) of the design, a set of transistors pairs (Ps) to specify matching transistors and an accompanying

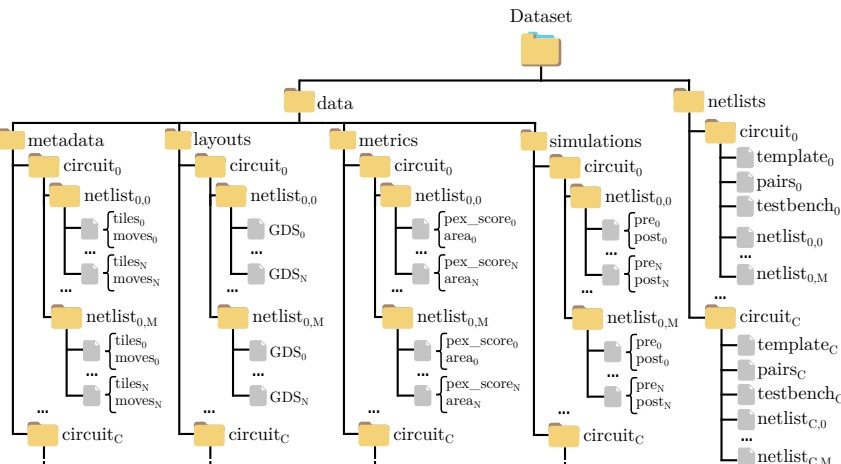

Figure 4: Dataset structure. It contains two directories, `netlists` and `data`. The latter is further divided into four subfolders, `simulations`, `metrics`, `layouts`, and `metadata`.

simulation testbench (`TB`) employed to verify the functionality and performance of the variants produced throughout the generation process. These elements, which the user provides, define both the design's logical behavior and evaluation criteria. *OSIRIS* generates as output a dataset comprising a layout (`GDS`) and quality of solution (`QoS`) files. `GDS` is a physical representation of the layout, while `QoS` is a report describing the corresponding layout.

**Fingers Permutation** The first stage, referred to as *Fingers Permutation*, receives as input `TP` and `Ps` and generates as output a set of netlists to be explored further in the pipeline. This stage introduces structural diversity by varying the number of fingers assigned to transistors in the circuit. Since finger count affects the layout topology and electrical characteristics such as parasitic capacitance and matching, this variation allows for exploring alternative physical implementations without altering the circuit's logical function. All valid combinations of finger assignments are enumerated, resulting in a total of `M` distinct permutations. A permutation of fingers is valid if it adheres *(i)* to matching requirements, for instance, current mirrors and differential pairs require their transistors to be as identical as possible, therefore to have the same number of fingers and *(ii)* the minimum gate dimension specified by the PDK in use. Each valid permutation is then used to annotate the `TP` to produce a new netlist $NL_i$ for $i = 0, \ldots, M$, thus generating a set of `M` netlists.

**Variants Generation** The *Variants Generation* stage processes each of the `M` generated netlists to generate a set of `GDS` and `QoS`. It takes as inputs the set of netlists and `TB`. This stage is further divided into two steps *(i) Baseline P&R* and *(ii) Random Exploration*. *Variants Generation* processes one netlist $NL_i$ at a time for $i = 0, \ldots, M$.

First, for a given $NL_i$, *Baseline P&R* performs a complete place-and-route flow to synthesize a baseline layout. Notably, *Baseline P&R* comprises a placer and a router. The placer encodes circuits using sequence-pairs (Murata et al., 2002), from which placement constraints are derived and solved with a geometric ILP formulation. Simulated annealing (Kirkpatrick et al., 1983) acts as a meta-heuristic, permuting the sequence pairs to explore alternative placements to minimize the half-perimeter wire length (HPWL). The routing process is split into two stages, global routing, performed with Dijkstra's algorithm (Dijkstra, 2022), and detailed routing, handled with the $A^*$ search algorithm (Hart et al., 1968) to resolve local congestion. The generated baseline layout is a faithful physical realization of $NL_i$. *Baseline P&R* integrates the place-and-route process along with the necessary steps to assess the validity of the generated layout, i.e., design rule checking (DRC), layout versus schematic (LVS), parasitic extraction (PEX), and simulation. This step produces three objects, a baseline layout $GDS_i^b$, the corresponding $QoS_i^b$, and an internal representation ($IR_i^b$), containing the coordinates (upper right and lower left corners) and type (nmos, pmos, capacitor, resistor) of each component of $NL_i$. The baseline plays a dual role in the dataset generation process. It is directly included in the final dataset as the baseline, i.e., variant b, representing an unmodified, post-P&R instance of the design. Moreover, it is the initial configuration from which all subsequent layout perturbations are derived.

Table 1: Dataset statistics. For each circuit, the number of explored netlists, the number of generated layout variants per netlist, the metrics, and the acquisition time are reported.

|  |  | Miller | Ahuja | Feed Forward | 5-Transistors | LPF |
|---|---|---|---|---|---|---|
| **Circuits components** | nmos | 5 | 10 | 7 | 3 | 5 |
|  | pmos | 4 | 4 | 6 | 2 | 3 |
|  | cap | 1 | 1 | - | - | 3 |
|  | res | 1 | - | - | - | 2 |
| **Explored netlist** |  | 146 | 143 | 224 | 129 | 229 |
| **Generated layouts per netlist** |  | 100 | 100 | 100 | 100 | 100 |
| **Total generated layouts** |  | 14 600 | 14 300 | 22 400 | 12 900 | 22 900 |
| **Mean and var. PEX score** (volts) |  | 0.0794 - 0.0007 | 0.1693 - 0.0005 | 0.1044 - 0.0012 | 0.0674 - 0.00005 | 0.3201 - 0.0036 |
| **Mean** ($\mu m^2$) **and var.** ($nm^2$) **area** |  | 1 748 - 628 | 1 776 - 753 | 768 - 778 | 414 - 3 314 | 5 401 - 1 530 |
| **Mean generation time (per layout)** [s] |  | 48 | 48 | 25 | 45 | 29 |
| **Total generation time** [h] |  | 195 | 191 | 156 | 162 | 187 |

Second, the *Random Exploration* step explores spatial variability. It receives as input $IR_i^b$ and produces as outputs a set of $N$ incremental variants each described by a $GDS_i^j$ and a $QoS_i^j$ for $j = 0, \dots, N$. Thus, iteration $j$ serves as baseline for iteration $j+1$, with $IR_i^0 = IR_i^b$. The *Random Exploration* is divided into three sub-steps, *(i) Shift (ii) P&R* and *(iii) Validate*, executed in a loop for $N$ successful iterations. *Shift* rolls two dice, one to choose which component to move and the other to choose the cardinal direction, i.e., up, down, left, right, to move the component towards. *Shift*, also, perturbates $IR_i^j$ to reflect the shifting. *P&R* reads $IR_i^j$, shifts the selected component accordingly, and, since the new placement invalidates the components connections, performs the routing of all netlist components considering the coordinates of the newly shifted component to ensure electrical connectivity, thus yielding $GDS_i^j$. Lastly, *Validate* assesses that the perturbed layout remains logically equivalent to the original netlist and estimates its behavior. To this end, the classic analog design flow performs DRC, LVS, PEX, and simulation. An iteration $j$ is successful if the *Validate* sub-step completes correctly. If it fails, the perturbation is reverted and the $j$-th iteration is repeated. At the conclusion of this process, the dataset contains a total of $M \times N$ variants. Each variant includes the layout in GDSII format ($GDS_i^j$) and a set of resources that reflects design quality in terms of physical and electrical metrics along with auxiliary metadata such as the specific netlist configuration and the nature of component displacements ($QoS_i^j$). Notably, to carry out LVS, PEX, and simulation, *OSIRIS* fully integrates with Netgen, Magic, and Ngspice, respectively, while DRC is respected by construction during place and route processes.

## 4 ANALOG LAYOUT DATASET

### 4.1 DATASET STRUCTURE

Figure 4 depicts the organization of files in the randomly generated dataset. It comprises all relevant information on each generated layout variant. It is divided into two main folders: `netlists` and `data`. The dataset was recorded on a Rocky Linux 8.10 machine equipped with an Intel Xeon 2.90GHz with 32 cores and 128GB of RAM. No GPU was used.

**Netlists** The `netlists` directory contains $C$ subfolders, one for each `circuit` class processed by *OSIRIS*. Each `circuit` subfolder refers to a specific analog circuit and comprises the circuit netlist `template`, which is a base netlist that serves as the canonical representation of the `circuit`, the `testbench`, in SPICE format, used to assess pre- and post-layout functionality, the `pairs` file reporting the matching transistors, and `M` netlist files in SPICE format. Each `netlist` inherits from the `template` information on component names, types, connectivity, and sizing while adopting a different permutation over the number of fingers. The `template` files are provided by the user while the sets of `netlist` files are the results of the *Fingers Permutation* stage of the *OSIRIS* as described in Section 3.2.

**Data** For each `circuit`, the `data` directory includes all necessary information to describe its corresponding design variations. This directory is organized into four subfolders, each containing relevant data for the `N` layout variants associated with every `netlist` of the given `circuit`:

Table 2: Analog datasets comparison. *OSIRIS* is the first framework aimed at back-end design that systematically explores and extensively covers the layout design space.

|  | CktGNN (Dong et al., 2023) | AnalogGenie (Gao et al., 2025a) | AMSNet (Tao et al., 2024) | ALIGN (Kunal et al., 2019) | **OSIRIS** |
|---|---|---|---|---|---|
| Design phase | Front-end | Front-end | Front-end | Back-end | Back-end |
| Items type | Netlists | Netlists | Netlists | Netlists | Layouts |
| Volume (#) | 60 000 | 3 350 | 824 | 23 | 87 100 |
| Physical validity check | N.A. | N.A. | N.A. | ✓ | ✓ |
| Parasitic awareness | ✗ | ✗ | ✗ | ✗ | ✓ |
| Data synthesis | ✓ | ✓ | ✗ | ✗ | ✓ |
| Extensibility | ✓ | ✓ | ✗ | ✗ | ✓ |

- `Simulations` comprises the pre- and post-layout simulation results for each layout variant, respectively in `pre` and `post` files. These files are obtained by testing each variant with the corresponding `circuit`'s `testbench`. Testbenches consist of AC small-signal simulations. The frequency is swept from 1 kHz to 1 GHz with 50 points per decade.
- `Metrics` captures the QoS associated with each layout variant in terms of physical and electrical properties. The metrics are *(i)* `pex_score` and *(ii)* `area`. The former quantifies the degradation in performance due to parasitic effects introduced during the layout process as a root mean square error between the `pre` and `post` results. At the same time, the latter measures the total silicon footprint of the layout. See Section 4.3 for a detailed description of the metrics. For both metrics, lower values indicate better performance.
- `Layouts` contains the physical layout files, `GDS`, generated for each variant in GDSII format. A `GDS` provides a precise geometric representation of the layout encoding polygons distributed across multiple metal layers, defining the physical implementation of the chip.
- `Metadata` contains the GDSII files for each individual component, `tiles`, and the accumulated movements, `moves`, performed on each tile up to iteration `j`-th.

Beyond serving as a large-scale physical-design benchmark, the *OSIRIS* dataset is explicitly structured for ML pipelines. Its paired pre- and post-layout simulations, detailed parasitic degradations, and spatial metadata enable tasks such as parasitic prediction, ML-guided placement, and component-level layout synthesis. The fine-grained organization of geometry, metrics, and traces provides supervision signals tailored to learning layout-dependent behaviors.

## 4.2 STATISTICS

Amplifiers and filters are widespread in analog design, serving as core components within a wide range of more complex systems, e.g., ADC/DAC and sensor interfaces (Franco, 2002; Huijsing, 2011; Carusone et al., 2011). Their widespread use and inherent component diversity, encompassing NMOS/PMOS transistors, resistors, and capacitors, as well as structural diversity with current mirrors and differential pairs, provide an ideal circuit template to investigate automatic layout flows. All circuits have been implemented in Skywater 130nm process development kit (PDK) (SKY) as it is mature, open-source, and well-integrated into available open-source CAD tools (Kahng, 2020; Herman et al., 2023; Chen et al., 2023b; Tsuchiya, 2024; Teo et al., 2025). Notably, a PDK is defined as a set of files and rules provided by a manufacturer to enable chip designers to create layouts of integrated circuits that comply with manufacturing capabilities, i.e., that can be taped out. Table 1 summarizes key statistics of the generated dataset, including the number and types (PMOS/NMOS transistors, resistors, and capacitors) of circuit components. It also reports the number of finger permutations, i.e., netlists, explored along with the number of layout variants produced per netlist and per circuit. Up to 229 finger permutations were explored for a single circuit, with 871 permutations evaluated across all circuits, resulting in 87 100 DRC-clean and LVS-verified layout variants. The generation of a single layout required an average of 39 seconds while the exploration of a single circuit required between 156 and 195 hours to complete. Table 2 compares the dataset generated by *OSIRIS* with state-of-the-art analog datasets. The comparison emphasizes its key contribution: providing a framework capable of generating extensive quantities of layout data specifically targeting the back-end phase of analog IC design. The dataset weighs 5 GB and required around 37 days to acquire. Appendix B illustrates the schematics of the four amplifier circuits used for dataset generation, along with samples of layout variants included in the dataset.

Figure 5: *RL Variants Generation* employs a two-level iterative RL-driven optimization process. It takes as inputs a set of netlists `NL` and a `TB` and generates as outputs a set of `GDS` and `QoS`. The outer level navigates transistor fingers permutations (stage *FinPerm Search*), while the inner level explores components movements (stage *Spatial Exploration*).

### 4.3 METRICS

**PEX Score**  When evaluating an analog layout, it is essential to validate the design's accuracy and ensure its performance reliability. A critical aspect of this evaluation involves accounting for parasitic elements, such as interconnect capacitances and resistances, which can significantly impact circuit behavior and degrade functionality. To quantify the influence of parasitics, a comparison between schematic-level (pre-layout) and post-layout simulation results is performed. This comparison is captured by the *pscore*, a metric that measures the discrepancy between pre- and post-layout outputs in volts (V). As defined in Equation 1, the pscore is computed as the RMSE between the pre-layout simulation results (pre) and the post-layout results (post):

$$pscore = \sqrt{\frac{1}{K} \sum_{i=1}^{K} (pre_i - post_i)^2} \tag{1}$$

where $K$ denotes the number of sampled points in each simulation trace. By construction, the pscore quantifies the impact of layout-induced parasitics, which degrade key analog performance metrics such as gain, bandwidth, noise, and stability. Its optimization is crucial for reliable circuit operation.

**Area**  In addition to electrical performance, physical layout characteristics, such as silicon area, are critical factors in analog circuit evaluation, particularly in cost-sensitive or space-constrained applications. The total layout area directly measures the silicon occupied by a design and is often used as a proxy for manufacturing cost and integration density. We compute the area of each generated layout as the summation of the areas of each component $t$, reported in Equation 2:

$$area = \sum_{t=0}^{T} W_t \cdot H_t \tag{2}$$

where $W_t$ and $H_t$ represent the width and height ($\mu m$) of the bounding rectangle enclosing the component $t$, and $T$ is the number of components. This metric reflects the overall footprint of the design and serves as an essential objective in layout optimization.

## 5 DESIGN SPACE EXPLORATION BASELINE

*OSIRIS* presents two main stages: *Fingers Permutation* and *Variants Generation*. The latter can implement different strategies to explore the layout design space for a given circuit. Section 3.2 presents a random exploration approach, while Section 5.1 introduces an RL-driven baseline methodology.

### 5.1 RL BASELINE METHODOLOGY

Figure 5 shows the *Variants Generation* implementation in a RL-driven iterative approach, i.e., *RL Variants Exploration*. It takes as inputs the set of netlists generated by *Fingers Permutation* and a `TB`. The output is a comprehensive set of `GDS` and `QoS` files. The implementation is based on two level of RL optimization loops, one for each degree of freedom. The outer one pivots around

searching the number of fingers permutations space while the inner one explores spatial movements for each component.

**Outer loop** The outer loop processes all netlists and the reward $R_i$ from the previous iteration to produce $N$ GDS and QoS files per netlist. It comprises four steps: *(i) FinPerm Search*, *(ii) Baseline P&R*, *(iii) RL Exploration*, and *(iv) Compute Reward*. *FinPerm Search* selects the next netlist $NL_i$ to explore and includes translating SPICE-formatted transistor finger permutations into RL-compatible inputs. Notably, *FinPerm Search* employs a fully connected agent that receives a vector containing the number of fingers for each transistor as input, and produces probabilities over possible finger values and maps them to categorical actions. It is trained via REINFORCE (Williams, 1992) as a straightforward, model-free policy gradient method. Appendix A details its architecture. *Baseline P&R* generates a baseline layout ($GDS_i^b$), quality metrics ($QoS_i^b$), and internal representation ($IR_i^b$). The *RL Exploration* step uses these as inputs and produces $N$ improved GDS and QoS variants for each baseline. Lastly, *Compute Reward* extracts $R_i$ from the set of QoS files as reported in Equation 3:

$$R_i = \max_{j=1,\dots N} \alpha \cdot (\text{pscore}_i^j - \text{pscore}_i^b) + \beta \cdot (\text{area}_i^j - \text{area}_i^b) \tag{3}$$

where $\text{area}_i^0 = \text{area}_i^b$ and $\text{pscore}_i^0 = \text{pscore}_i^b$ while $\beta$ and $\alpha$ are tuning parameters. Notably, if the outer loop's agent chooses an invalid finger configuration, a negative reward is fed to it and a new choice is made.

**Inner loop** The *RL Exploration* is the inner loop, depicted in Figure 5. It mirrors the *Random Exploration* strategy (Section 3.2). It takes as input $IR_i^b$ and outputs $N$ GDS and QoS files for each baseline, which are passed to the outer loop for further processing and storing. *RL Exploration* comprises three steps: *(i) RL Place*, *(ii) P&R*, and *(iii) Validate*. *RL Place* uses an actor-critic agent operating on an input vector that encodes each component's type and coordinates. The agent includes an initial shared portion which splits into two branches *(i)* actor and *(ii)* critic, providing the action to move a component and the state value of the current iteration $IR_i^j$. The inner agent is trained with Proximal Policy Optimization (cPPO) (Schulman et al., 2017), which is well-suited for handling discrete action spaces. Appendix A details its architecture. *P&R* implements the action by shifting the selected component and performing components routing. *Validate* ensures adherence to netlist $NL_i$ and assesses functionality. Moreover, *Validate* computes the agent's reward ($r_j$) as reported in Equation 4:

$$r_j = \alpha \cdot (\text{pscore}_i^j - \text{pscore}_i^b) + \beta \cdot (\text{area}_i^j - \text{area}_i^b) \tag{4}$$

where $\alpha$ and $\beta$ are tuning parameters. The sub-steps of *RL Exploration* are performed in a loop for $N$ iterations. Opposite to *Random Exploration*, if an iteration is unsuccessful, it is not repeated, the episode ends, and a penalty reward is fed to the inner agent.

# 6 OSIRIS DATASET USE CASE

This section illustrates a representative use case enabled by the *OSIRIS*-generated open-source dataset described in Section 4, focusing on automated component-level layout generation. In analog back-end design, each circuit layout is constructed from the physical layouts of its constituent components. Layouting each component instance of a circuit is a time-consuming and error-prone task that requires PDK-specific knowledge. To this end, the *OSIRIS*-generated dataset is used to fine-tune an LLM-based generator to deliver DRC-free and LVS-verified layouts at component level. The use case leverages the *OSIRIS*-generated dataset to fine-tune the Qwen3-14B (Yang et al., 2025) LLM to generate DRC-free, LVS-verified capacitor layouts in SkyWater $130\,nm$ directly from sizing targets. The model is fine-tuned using low rank adaptation (LoRA) (Hu et al., 2021) and evaluated on test samples to verify dimensional correctness and syntactic validity. Fine-tuning involved $10\,000$ samples (split between training, validation, and test sets) extracted from the `primitives` of the *OSIRIS*-generated dataset. Fine-tuning employed a H100 GPU equipped with 96 GB of dedicated RAM, rented specifically for this task. The fine-tuned model produces 100% valid outputs and perfect dimensional fidelity, whereas the vanilla Qwen3-14B model lacks any geometric competence and fails to produce valid layouts. Appendix C provides in-depth details on data pre-processing, fine-tuning hyperparameters selection, and prompt and generated capacitor layout examples.

Table 3: Results comparison between ALIGN (Kunal et al., 2019), MAGICAL (Xu et al., 2019), the random baseline (Section 3), and the RL-driven methodology (Section 5). Each circuit block spans three rows showing pscore (volts), area ($\mu m^2$), and acquisition time (hh:mm:ss).

| Circuit | MAGICAL | | | ALIGN | | | Random | | | RL | | |
|---|---|---|---|---|---|---|---|---|---|---|---|---|
| | pscore ($\downarrow$) (V) | area ($\downarrow$) ($\mu m^2$) | time (hh:mm:ss) | pscore ($\downarrow$) (V) | area ($\downarrow$) ($\mu m^2$) | time (hh:mm:ss) | pscore ($\downarrow$) (V) | area ($\downarrow$) ($\mu m^2$) | time (hh:mm:ss) | pscore ($\downarrow$) (V) | area ($\downarrow$) ($\mu m^2$) | time (hh:mm:ss) |
| Miller | 0.2739 | 1 980 | 00:02:05 | 0.142 | 1 836 | 00:01:08 | 0.0012 | 1 770 | 195:00:00 | 0.00069 | 1 733 | 96:00:00 |
| Ahuja | 0.5184 | 2 190 | 00:01:57 | 0.315 | 1 906 | 00:01:08 | 0.120 | 1 805 | 191:00:00 | 0.120 | 1 797 | 70:00:00 |
| Feed Forward | 0.2087 | 798 | 00:01:20 | 0.210 | 806 | 00:01:18 | 0.037 | 809 | 156:00:00 | 0.024 | 768.5 | 62:00:00 |
| 5-Transistors | 0.2554 | 347 | 00:01:03 | 0.093 | 183 | 00:00:48 | 0.050 | 266.3 | 162:00:00 | 0.047 | 444.5 | 50:00:00 |
| LPF | - | - | - | 0.501 | 8 430 | 00:01:20 | 0.102 | 7 916 | 187:00:00 | 0.064 | 6 635 | 67:00:00 |

# 7 EXPERIMENTAL RESULTS

To evaluate the benefits of integrating learning-driven strategies within *OSIRIS*, the RL-based exploration method (Section 5) is compared against *(i)* state-of-the-art single-pass layout generators MAGICAL and ALIGN, and *(ii)* the random exploration strategy inherent to *OSIRIS* (Section 3). The comparison's aim is to show that when *OSIRIS* is paired with different exploration approaches, including RL, it can produce layouts with improved parasitic metrics and area compared to single-pass baselines. MAGICAL and ALIGN provide meaningful references as they represent the most capable open-source automatic back-end flows currently available. Two NVIDIA GeForce GTX 1080Ti GPUs were used for the RL-related experiments. Appendix A details hyperparameter selection. Table 3 presents the comparative analysis of the four approaches. The evaluation considers three metrics for each circuit benchmark: pscore, layout area, and total acquisition time. The RL-based approach achieves lower pscore values across all benchmarks, suggesting improved handling of parasitic effects in the resulting layouts. For the Miller and Feed Forward amplifiers, in particular, the RL method reports pscore values more than an order of magnitude smaller than those produced by ALIGN and MAGICAL, indicating that learning-guided placement can be beneficial when integrated with *OSIRIS*-generated layouts. Regarding layout area, the RL baseline usually yields comparable or smaller designs. For example, the Miller and Feed Forward amplifiers produce the lowest pscore and the most compact layouts. For the Ahuja amplifier, although RL and Random yield the same pscore, the RL-based layout occupies slightly less area (1 797 $\mu m^2$ vs. 1 805 $\mu m^2$). One exception is the 5-Transistors OTA, where the RL approach results in a larger layout area (444.5 $\mu m^2$) than baselines. However, it still attains a lower pscore, indicating a trade-off favoring electrical performance over area reduction. The RL-based exploration strategy remains effective across different circuit families. For instance, on the low-pass filter (LPF) benchmark, it attains both the smallest layout area and the lowest pscore. Notably, MAGICAL failed to generate any valid LPF layouts. The RL-based method also demonstrates reduced acquisition time compared to the Random baseline due to more efficient exploration of the solution space, converging faster towards high-quality solutions. For example, in the Miller and Ahuja circuits, it achieves nearly a 50% reduction in runtime. Across all benchmarks, it reports the shortest acquisition times. These results indicate that reinforcement learning can be a viable strategy for efficient and performance-aware analog layout space exploration.

# 8 CONCLUSIONS

This work presents *OSIRIS*, an end-to-end back-end framework for generating large quantities of DRC-clean, LVS-verified analog layouts to advance ML research in circuit design. *OSIRIS* enables scalable creation of annotated datasets and provides a public release of 87 100 layout variants across four designs. An RL baseline demonstrates its use for optimizing functional and non-functional metrics, such as parasitics and area. Unlike prior datasets, *OSIRIS* is open-source, configurable, and tailored for ML-driven optimization. While the current release targets five circuits in a single technology node, the *OSIRIS* pipeline is inherently extensible. Ongoing development includes support for additional circuit families, richer perturbation operators, and transfer-learning functionality to layout the same circuit targeting different PDKs. By providing a modular and open-source foundation for iterative parasitic-aware exploration, *OSIRIS* lays the groundwork for future learning-driven analog back-end flows.

## 9 ETHICS STATEMENT

Research in analog design automation has the potential to indirectly impact a wide range of industries, including healthcare, autonomous systems, and communications. By accelerating the development of high-performance, energy-efficient analog ICs, frameworks such as *OSIRIS* may contribute to faster innovation in medical devices, sensing platforms, and safety-critical applications like autonomous driving. While these applications are broadly beneficial, they also raise ethical considerations. Improved design productivity could lower entry barriers, enabling broader access to advanced technologies, but it may also be leveraged in sensitive domains (e.g., surveillance or military systems). We emphasize that *OSIRIS* is released as an open, transparent research tool intended to foster reproducibility, fairness, and broad participation in the ML-for-EDA community. Responsible use should remain a guiding principle, particularly in safety-critical or high-stakes domains.

## 10 REPRODUCIBILITY STATEMENT

*OSIRIS* is released at the anonymous HuggingFace repository link mentioned in Section 1. It includes *(i)* *OSIRIS* end-to-end pipeline delivering DRC-free and LVS-verified layouts *(ii)* random exploration strategy, and *(iii)* RL-driven exploration strategy. The random exploration strategy is described in Section 3 while the RL-driven one is detailed in Section 5. Moreover, the repository contains instructions on how to install and run the code. In addition to the code, a dataset generated by the random exploration strategy is also released at the same HuggingFace repository mentioned in Section 1. Section 4 provides an in-depth description of the dataset. Appendix A provides implementation details regarding the RL agents, while Appendix B reports the schematics of the analyzed circuits and samples of layouts released in the dataset.

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

# A AGENTS AND HYPERPARAMETERS

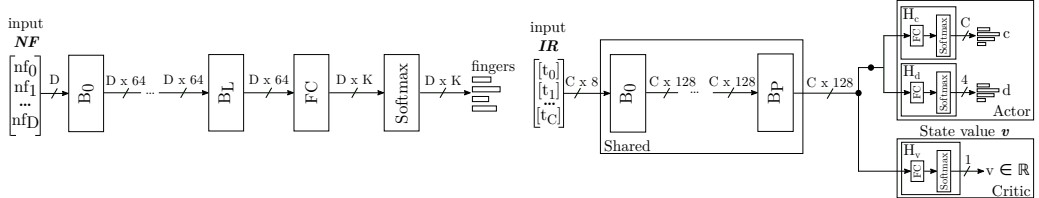

(a) Outer agent exploring the permutations of number of fingers. Part of *FinPerm Search* step.

(b) Inner agent exploring components movement within their halo. Part of *Shift* sub-step of *RL Exploration* stage.

Figure 6: Agent architectures employed in the two-level iterative optimization process of *RL Variants Generation*. (a) explores the different permutations of the number of fingers for each transistor and (b) explores the components' movement inside their halo.

Figure 6 shows the two agents' architectures used in the two-level iterative optimization process presented in Section 5.1.

In particular, Figure 6a shows the outer loop agent, part of the *FinPerm Search* step. It receives as input a vector (*NF*) with the number of fingers (*nf*) for each transistor at iteration i-1, outputs probabilities over possible finger values, and maps them to discrete choices to assign updated counts. The network consists of blocks (B) made of a fully connected layer (FC) with ReLU activation, followed by an FC layer and softmax. Here, *D* is the number of transistors in the netlist and *K* is the number of allowable finger values for the given TP.

Figure 6b reports the inner agent, part of the *Shift* sub-step of *RL Exploration*. It determines the action (*A*) by selecting a component (c) and a movement direction (d), and estimates the state value (*v*) of $\text{IR}_i^j$. It operates on an input vector (*IR*) encoding each component's type (one-hot) and coordinates, yielding an $8 \times C$ array. The agent adopts an actor-critic structure with *Shared*, *Actor*, and *Critic* modules. *Shared* consists of B blocks defined as in the outer agent. Actor has two heads, $H_c$ for component selection and $H_d$ for direction (up, down, right, left), each implemented as one FC layer with softmax, jointly defining *A*. Critic includes a head ($H_v$) outputting *v*, the estimated value of *IR*.

*RL Place* uses an actor-critic agent, see Figure 6b, operating on an input vector *IR* that encodes each component's type (one-hot) and coordinates, yielding 8 features per component. The agent includes *Shared P FCs* layers, and two branches: *Actor*, with softmax heads $H_t$ and $H_{dir}$ selecting a component and movement direction (defining action *A*), and *Critic*, producing a scalar *v* estimating the value of *IR*.

Table 4: Outer agent.

| Parameter | Value |
|---|---|
| K | {2, 4, 6, 8, 10, 12, 14, 16} |
| L | 5 |
| Units per B block | $64 \times D$ |
| D | {5, 9, 13, 14} |
| Learning rate | 1e-5 |
| Optimizer | Adam |
| Reward penalty | -10 |

Table 5: Inner agent.

| Parameter | Value |
|---|---|
| P | 5 |
| Units per B block | $128 \times C$ |
| C | {5, 11, 13, 15} |
| $\gamma, \epsilon$ | 0.99, 0.2 |
| Learning rate | 1e-5 |
| Optimizer | Adam |
| Entropy coeff. | 0.01 |
| Batch size | 16 |
| Replay memory size | 128 |
| Reward penalty | -0.001 |

Table 4 and Table 5 detail the parameters used to design and train both agents. In particular, *D* is the number of transistors while *C* is the total number of components in each circuit, respectively, 5-Transistors, Miller, Ahuja, and Feed Forward. Notably, $\alpha = 5$ and $\beta = 1.5$ to prioritize pscore improvements. The shifting amount is fixed to 100nm.

## B  CIRCUITS SCHEMATICS AND LAYOUT SAMPLES

Figure 7 reports the schematics of the four amplifier circuits employed throughout this work, while Figure 8, Figure 9, Figure 10, and Figure 11 show two representative layout variants, generated by *OSIRIS* and included in the released dataset, for Miller, Ahuja, Feed Forward, and 5-Transistors circuits respectively.

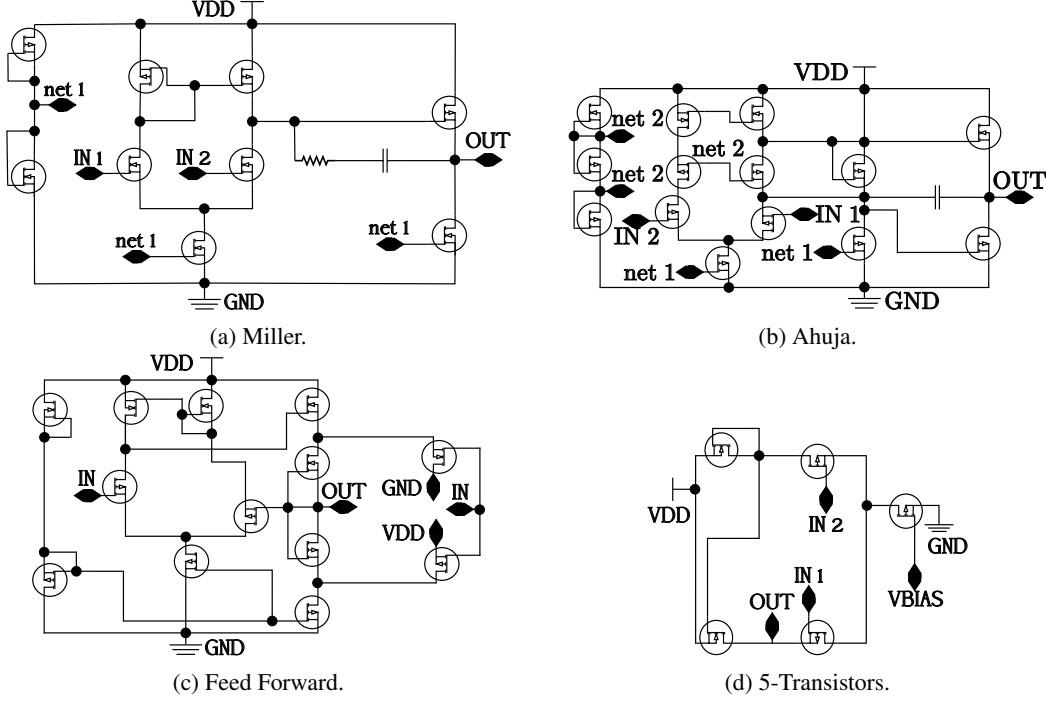

(a) Miller.

(b) Ahuja.

(c) Feed Forward.

(d) 5-Transistors.

Figure 7: Schematics of the four representative circuits explored using *OSIRIS*.

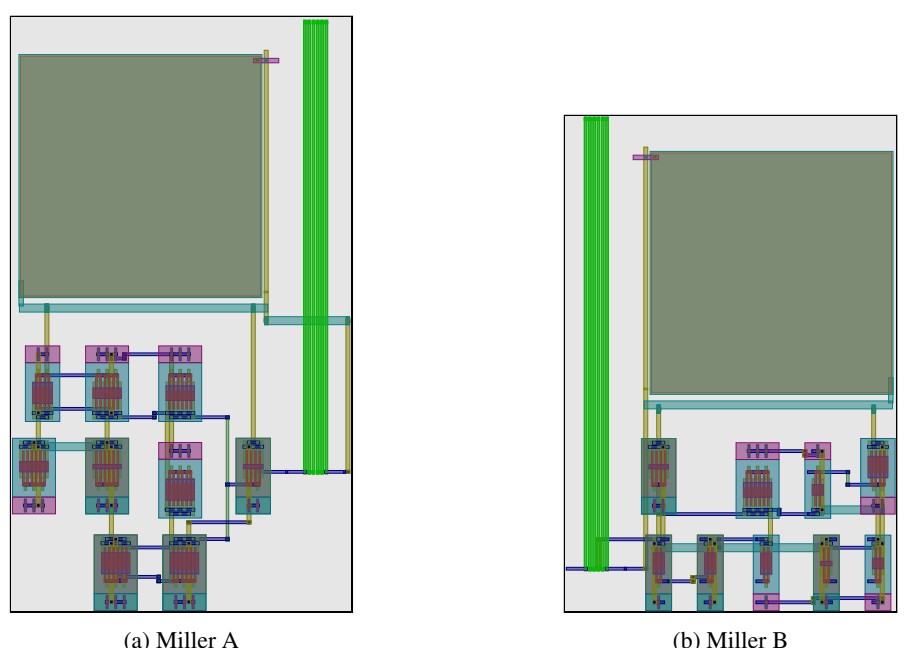

(a) Miller A

(b) Miller B

Figure 8: Examples of Miller layout variants generated by *OSIRIS*.

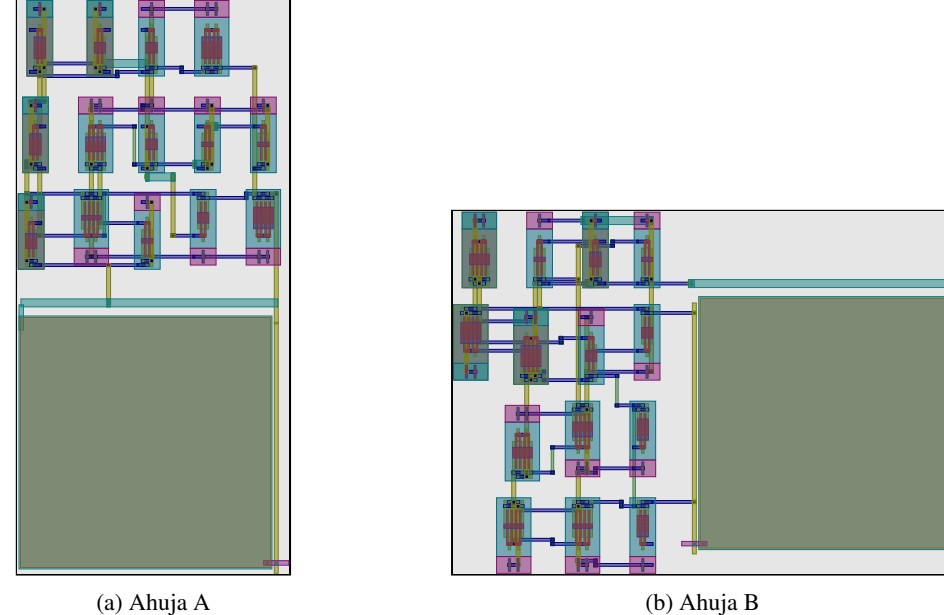

(a) Ahuja A
(b) Ahuja B

Figure 9: Examples of Ahuja layout variants generated by *OSIRIS*.

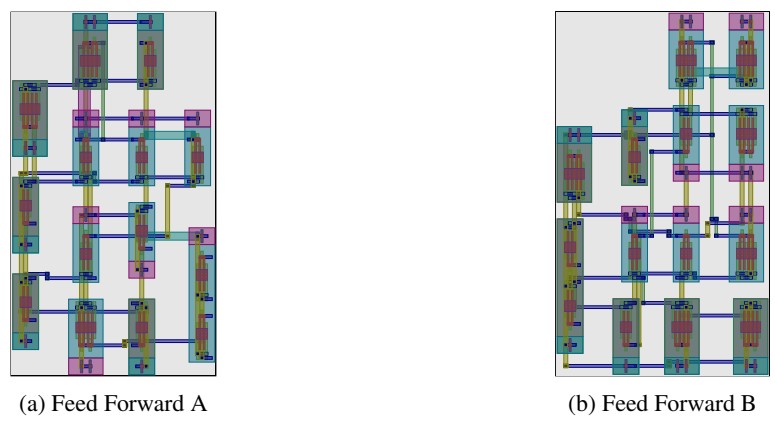

(a) Feed Forward A
(b) Feed Forward B

Figure 10: Examples of Feed Forward layout variants generated by *OSIRIS*.

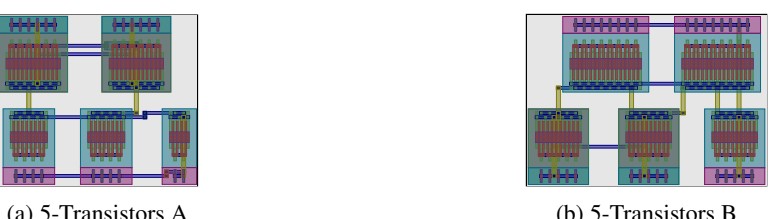

(a) 5-Transistors A
(b) 5-Transistors B

Figure 11: Examples of layouts generated by *OSIRIS* and included in the released dataset.

Table 6: Qwen3-14B fine-tuning configuration.

| Component | Setting |
|---|---|
| Base model | Qwen3-14B |
| Quantization | 4-bit NF4 (QLoRA) |
| LoRA rank $r$ | 64 |
| LoRA $\alpha$ | 16 |
| LoRA target modules | Attention and MLP projections |
| Trainable params | $\approx 67M$ ($0.48\%$ of total) |
| Optimizer | AdamW |
| Learning rate | $2 \times 10^{-4}$ |
| Betas | $(0.9,\ 0.999)$ |
| Weight decay | 0.01 |
| Scheduler | Cosine with warmup |
| Warmup steps | 1,000 |
| Batch size (per device) | 4 samples |
| Gradient accumulation | 1 step |
| Max sequence length | 4,096 tokens |
| Epochs | 3 |
| Loss | Next-token cross-entropy (completion-only) |

## C  LLM FINE-TUNING

### C.1  FINE-TUNING SETUP

Table 6 details the setup to fine-tune Qwen3-14B to generate DRC-free and LVS-verified layouts of capacitor components. In particular, the fine-tuning dataset comprises $10\,000$ capacitor layouts extracted from the `primitives` of the open-source released dataset, with $90\%$ training, $5\%$ validation, and $5\%$ testing splits. The model uses 4-bit quantization and LoRA-64. The fine-tuning process took 3 epochs on a H100 GPU equipped with 96 GB of RAM.

### C.2  DATA PRE-PROCESSING

The raw dataset comprises approximately $10\,000$ capacitor designs, each represented by two complementary files: *(i)* a SPICE netlist that specifies electrical parameters (length and width in $\mu m$), and *(ii)* a GDS layout file that encodes the physical geometry as a collection of polygons distributed across process-specific layers. To make these heterogeneous data suitable for sequence-based fine-tuning, we apply a systematic pre-processing pipeline. First, we parse the SPICE netlist to extract target dimensions and derive the design capacitance using technology-specific model files. Next, we load the corresponding GDS file and convert each of its polygon, regardless of its original vertex count, into its axis-aligned bounding box, represented as a six-element tuple: ($[layer$, $datatype$, ($x_{min}$,$y_{min}$,$x_{max}$,$y_{max}$]). To ensure uniform input dimensions, each layout is padded to a fixed maximum length, filling missing entries with zeros. Spatial coordinates are, then, normalized to stabilize gradients. Finally, structured prompts are constructed by pairing each layout's specifications (capacitance, length, width, and polygon budget) with its corresponding JSON-encoded polygons sequence. This end-to-end transformation converts raw data and netlists into a numerically conditioned, token-based representation that the LLM can learn to map from specifications to manufacturable 2D geometries.

### C.3  PROMPTS EXAMPLES

Figure 12 presents the prompt used to query the fine-tuned Qwen3-14B model to synthesize the layout of a capacitor with target capacitance $200pF$ and geometry $10\mu m \times 10\mu m$. The corresponding output is also shown: for each polygon in the generated layout, the model produces the fabrication layer and the associated coordinates, which together define a complete, manufacturable capacitor layout. For comparison, Figure 13 illustrates the prompt used to query the not fine-tuned Qwen3-14B model. Although the prompt is deliberately more structured to guide the model, the base LLM

---

**Fine-tuned Qwen3-14B prompt and response**

You are Khufu, an analog layout assistant. Given the capacitor targets, reply with strict JSON: {"polygons": [[layer, datatype, xmin, ymin, xmax, ymax], ...]}. Each polygon is a 6-element array.
Only output JSON and stay within the canonical layer/datatype pairs.
Input:
- capacitance_f: 2.000000e-10
- length_m: 1.000000e-05
- width_m: 1.000000e-05
- polygon_budget: 250

- - - - - - - - - - - - - - - - - - - - - - - - - - - - - - - - - - - - - - - - - - - -

Response:
{"polygons": [["metal1", 0, 0, 0, 1e-5, 1e-5], ..., ["metal1", 0, 1e-5 + 1e-7, 1e-5, 2e-5 + 1e-7, 2e-5 + 1e-7]]}

Figure 12: Example of prompt, and corresponding response, used to query the layout of a capacitor given the required values of capacitance, width, and length.

lacks the PDK-specific geometric knowledge that the fine-tuned version acquires from the OSIRIS dataset, and therefore cannot generate valid physical layouts.

## C.4 GENERATED CAPACITORS LAYOUT EXAMPLES

Figure 14 compares the layouts produced by the fine-tuned and vanilla Qwen3-14B models for a $10\mu m \times 10\mu m$ capacitor. The fine-tuned model generates a physically valid structure: two correctly overlapping metal plates forming the parallel-plate capacitor, along with well-positioned connection pins, yielding a DRC-clean and LVS-consistent layout. In contrast, the vanilla one fails to generate the required plate overlap and instead outputs disjoint shapes that do not satisfy the geometric and electrical requirements of a functional capacitor.

---

**Vanilla Qwen3-14B prompt and response**

You are Khufu, an expert analog layout assistant specialized in generating capacitor layouts. Your task is to generate a complete layout in strict JSON format based on the given capacitor specifications.
CRITICAL: Layer/Datatype Requirements
You MUST generate polygons ONLY using these specific layer/datatype pairs:
- Layer 89, Datatype 44: Active_Area (main capacitor body)
- Layer 235, Datatype 0: Device_Boundary (alternative 1)
- Layer 235, Datatype 4: Device_Boundary (alternative 2)
- Layer 70, Datatype 20: Device_Boundary (alternative 3)
- Layer 71, Datatype 20: Terminal_1 and Terminal_2
- Layer 70, Datatype 44: Contacts
DO NOT use any other layer/datatype combinations. These are the ONLY valid pairs.
### Geometric Validity Constraints (MANDATORY):
You must ensure the following spatial relationships:
1. **Terminal_1** (layer 71, datatype 20) must be FULLY CONTAINED within **Active_Area** (layer 89, datatype 44)
- This means: Terminal_1.xmin ¿= Active_Area.xmin AND Terminal_1.xmax ¡= Active_Area.xmax
- AND: Terminal_1.ymin ¿= Active_Area.ymin AND Terminal_1.ymax ¡= Active_Area.ymax
2. **Terminal_2** (layer 71, datatype 20) must be FULLY CONTAINED within **Device_Boundary** (use ONE of: layer 70/datatype 20, OR layer 235/datatype 0, OR layer 235/datatype 4)
- Same containment rule applies
3. **All contacts** (layer 70, datatype 44) must be FULLY CONTAINED within their respective terminals (layer 71, datatype 20)
4. **Containment formula**: For any child polygon inside a parent polygon:
child.xmin >= parent.xmin AND
child.ymin >= parent.ymin AND
child.xmax <= parent.xmax AND
child.ymax <= parent.ymax
### Layout Structure Requirements:
Your layout MUST include AT MINIMUM:
- 1 polygon for Active_Area (89,44) - the main capacitor structure
- 1 polygon for Device_Boundary (choose from 70,20 or 235,0 or 235,4)
- 2 polygons for Terminals (71,20) - one for each terminal
- Multiple polygons for Contacts (70,44) - distributed across both terminals
### Output Format (STRICT JSON):
```json
"polygons": [
[layer, datatype, xmin, ymin, xmax, ymax],
[layer, datatype, xmin, ymin, xmax, ymax], ...]
Each polygon is a 6-element array: [layer, datatype, xmin, ymin, xmax, ymax]
Coordinates are in micrometers ($\mu m$)
xmax must be ¿ xmin
ymax must be ¿ ymin
Design Flexibility:
You have creative freedom in:
Terminal placement (as long as containment rules are satisfied)
Contact arrangement and distribution
Number of contacts (within polygon_budget)
Polygon positioning and sizing
Choice of Device_Boundary layer variant (70,20 or 235,0 or 235,4)
Input Specifications:
capacitance_f: [TARGET_CAPACITANCE] (in Farads, e.g., 1.5e-15)
length_m: [TARGET_LENGTH] (in meters, e.g., 2.0e-5)
width_m: [TARGET_WIDTH] (in meters, e.g., 1.5e-5)
polygon_budget: (maximum number of polygons)
IMPORTANT REMINDERS:
- Use ONLY the 6 specified layer/datatype pairs
- Verify ALL containment constraints before finalizing
- Output ONLY valid JSON with no additional text
- The Active_Area (89,44) dimensions should approximate the target length × width
- All coordinates must be positive and properly ordered (min ¡ max)
Required design layout as INPUT:
- capacitance_f: 4,050000e-09
- length_m: 4.500000e-05
- width_m: 4.500000e-05
- polygon_budget: 500

- - - - - - - - - - - - - - - - - - - - - - - - - - - - - - - - - - - - - - - - - - - - - -

Response:
"polygons": [[89, 44, 0.0, 0.0, 10.0, 10.0], [70, 20, 0.0, 0.0, 10.0, 10.0], [71, 20, 2.0, 8.0, 8.0, 10.0], [71, 20, 2.0, 0.0, 8.0, 2.0], [70, 44, 2.5, 8.5, 3.5, 9.5], [70, 44, 4.5, 8.5, 5.5, 9.5], [70, 44, 6.5, 8.5, 7.5, 9.5], [70, 44, 2.5, 0.5, 3.5, 1.5], [70, 44, 4.5, 0.5, 5.5, 1.5], [70, 44, 6.5, 0.5, 7.5, 1.5], [235, 4, 0.0, 0.0, 10.0, 10.0]]

Figure 13: Example of prompt, and corresponding response, used to query the layout of a capacitor given the required values of capacitance, width, and length.

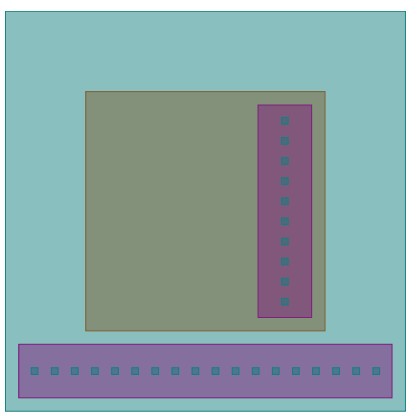
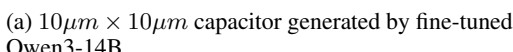
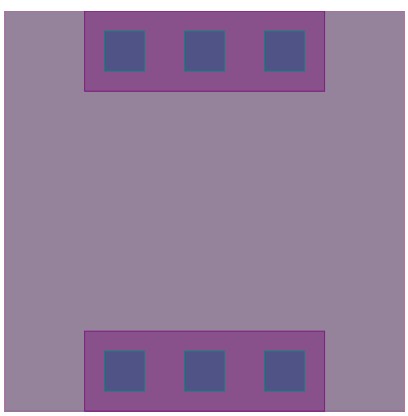

(a) $10\mu m \times 10\mu m$ capacitor generated by fine-tuned Qwen3-14B

(b) $10\mu m \times 10\mu m$ capacitor generated by not fine-tuned Qwen3-14B

Figure 14: Examples of capacitor layouts generated by (a) fine-tuned Qwen3-14B LLM and (b) not fine-tuned Qwen3-14B.

