# OpenReview forum: "OSIRIS: Bridging Analog Circuit Design and Machine Learning with Scalable Dataset Generation"
_ICLR.cc/2026/Conference — ICLR 2026 Poster_

### Official Review · Reviewer_E2BV · 2025-10-16

**Soundness:** 4
**Presentation:** 4
**Contribution:** 4
**Rating:** 8
**Confidence:** 5

**Summary:**

This work presents a synthetic pipeline that can generate large volumes of analog circuit layouts. This work also presents an RL design framework that can optimize circuit performance based on post-layout performance and show competitive performance compared to early SOTA MAGICAL and ALIGN. The entire framework will be open-sourced.

**Strengths:**

1. This work addresses an important problem for analog circuit design automation and provides an invaluable contribution
While the existing analog circuit dataset only contains basic netlists, this work is able to provide its layout, which is essential for determining the circuit's real-world performance. The entire synthetic pipeline is automated and can address the data shortage issue faced by the entire analog circuit design flow (front-end and back-end). I believe this work will drive the entire analog EDA filed forward.
2. Excellent presentation and writing
3. Strong experiment results and supplement material support
The attached anonymous link contains all the material needed for determining this work's reproducibility.

**Weaknesses:**

1. More examples and results beyond the op-amp can further strengthen this paper

**Questions:**

see weakness

---

> ### Author Response · Authors · 2025-11-24
>
> We thank the reviewer for the positive assessment and for recognizing the contribution of OSIRIS to advancing analog design automation. We appreciate the reviewer’s suggestion regarding broader circuit coverage, and we agree that extending beyond operational amplifiers further strengthens the paper.
>
> The choice of four amplifier topologies stems from their central role in analog design and their structural diversity, which includes current mirrors, differential structures, and multiple component types. These features make them an effective foundation for validating the first end-to-end, ML-compatible analog layout generation pipeline. Nonetheless, OSIRIS was designed with modularity in mind: the netlist templating mechanism, placement and routing engine, and verification step allow straightforward extension to additional circuit classes.
>
> To better illustrate this generality and within the review period, we added a low-pass filter (LPF) example to the revised manuscript. This example demonstrates how OSIRIS can accommodate circuits that differ structurally from amplifiers, confirming that the framework can be used to generate layouts and metadata for a wider set of analog building blocks within a relatively short amount of time. We emphasize that ongoing work is expanding the library of supported circuits.

---

> > ### Comment · Reviewer_E2BV · 2025-11-25
> >
> > I acknowledge the author's thoughtful rebuttal. My concern about limited circuit type coverage has been addressed. I would like to maintain my score.

---

### Official Review · Reviewer_z7C5 · 2025-10-30

**Soundness:** 3
**Presentation:** 3
**Contribution:** 3
**Rating:** 4
**Confidence:** 3

**Summary:**

In order to address the issue of the lack of open-source high-quality datasets in the field of analog integrated circuits(IC), this paper presents a pipeline called OSIRIS, which is an end-to-end back-end framework for generating large quantities of analog layouts to advance ML research in circuit design. At the same time, it provides a public release of 64200 layout variants across four designs and an RL baseline.

**Strengths:**

1. Systematically generated a full-link dataset covering 4 types of typical amplifiers and 64000+ samples. All samples passed the “Design Rule Check” and “Layout-Circuit Consistency Check”, ensuring the “industrial-grade reliability” of the data. It is the first publicly available, reproducible, and fully annotated large-scale simulation IC layout dataset.

2. The process and methods have strong scalability. The automated processing flow has potential to handle other types of analog circuits and advanced manufacturing processing, without requiring significant modifications.

3. The supporting methods is highly consistent with the conference theme. The proposed two-level RL optimization framework  integrates machine learning technology deeply into the post-synthesis design process of analog ICs. It not only reflects the core direction of "machine learning-driven electronic design automation (EDA)", but also provides method validation for the practical value of the dataset, and is highly consistent with the positioning of ICLR, which focuses on innovative applications of machine learning.

**Weaknesses:**

1. Only covering 4 types of amplifiers and 130nm process, the circuit and process coverage is insufficient.

2. The experimental part of the main text does not fully elaborate on the quality comparison of the data set and the model effect based on it, to prove the lightweighting. Without conducting verification in conjunction with specific simulation IC design tasks, there is a lack of quantitative results demonstrating the improvement in model performance of this dataset in actual tasks. As a result, the practical value and superiority of the dataset have not been fully verified through experiments, and the argumentation is not very persuasive.

3. Based on the data, the generation method is inefficient. It fails to meet the requirements of "millions of samples" in machine learning research or the "rapid iterative design" scenarios in the industrial sector. However, the article does not deeply analyze the core sources of the time-consuming bottlenecks, nor does it propose targeted optimization solutions.

**Questions:**

1. The dataset only covers 4 types of amplifiers. How can it be demonstrated that it has generalization capabilities for other analog circuits  and advanced processes? And It is advisable to verify the advantages of the dataset in some specific tasks.

2. The process of generating the dataset takes a considerable amount of time (more than 50 hours for a single circuit). Is it practical for large-scale machine learning tasks? Are there any optimization plans?

---

> ### Author Response · Authors · 2025-11-24
>
> We thank the reviewer for the constructive feedback and for highlighting the value of releasing a large-scale, fully verified analog-layout dataset together with a machine-learning–driven back-end optimization framework.
>
> We acknowledge that the current release includes four amplifier topologies implemented in the SkyWater 130 nm PDK. However, we note that the complexity of obtaining DRC- and LVS-clean layouts prevents the availability of open-source layout datasets, thus making our dataset extremely valuable to investigate novel ML-driven layout strategies. These circuits were chosen because they represent fundamental analog building blocks and incorporate heterogeneous devices, current mirrors, and differential structures that frequently recur in higher-level systems. Their diversity provides a relevant testbed to validate the first end-to-end, ML-compatible back-end flow of this kind. Importantly, OSIRIS is designed to be technology- and topology-agnostic: its modular pipeline, based on a parameterized netlist template, a PDK-aware placer/router, and fully automated verification steps, can be applied to new circuits and advanced nodes without conceptual modification. We are actively extending it to additional circuit classes and technology nodes. To this end, Table 3, in Section 7 “Experimental Results”, has been enriched with a new circuit topology, namely low-pass filter (LPF), showcasing a different circuit family other than amplifiers.
>
> Regarding the experimental validation of dataset usefulness, we agree that demonstrating how the dataset improves the performance of downstream ML tasks strengthens the paper. We clarify that the dataset enables practical ML scenarios such as performance-prediction training, ML-guided placement scoring, or automatic generation of device layouts.  To this end, the revised manuscript includes a concrete demonstration showing how OSIRIS-generated samples can be used to fine-tune an LLM to generate GDS layouts of components, in particular, focusing on capacitors generation. Details are described in Section 6 “OSIRIS Dataset Use Case” and Appendix C “LLM Fine-tuning”.
>
>  The goal of Section 5 “Design Space Exploration Baseline” along with Section 6 “OSIRIS Dataset Use Case” and Appendix C “LLM Fine-tuning” is to showcase the different uses of OSIRIS which can be leveraged (i) as a backbone for layout design space exploration methodologies and (ii) to generate large volumes of data exhaustively detailed to support task-specific models training.
>
> On concerns related to generation efficiency, we agree that runtime is an important factor for both ML-scale datasets and industrial prototyping scenarios. OSIRIS aims to guarantee physical correctness (DRC-free and LVS-verified layouts) and parasitic accuracy, which necessitates full verification and simulation for each sample. Despite this, the pipeline generates a complete, fully verified layout roughly every 50 seconds on a 32-core CPU machine for a Miller amplifier, which is competitive given the complexity of analog verification. Moreover, OSIRIS is intended to produce data with high physical fidelity rather than synthetic abstractions. While this results in higher per-sample cost, it enables ML research that explicitly incorporates routing-dependent parasitics and layout-accurate features, capabilities that front-end datasets cannot provide. Furthermore, the acquisition time reported in the paper reflects a single-machine setup, however, OSIRIS inherently allows parallelization at the layout generation level, therefore compute power scaling can substantially improve dataset generation time. As computing resources scale, and as additional optimizations are integrated into the placer, router and verification stages, generating larger datasets will become increasingly practical.
>
> Concerning generalization to larger circuits and more advanced processes, OSIRIS is intentionally built around abstractions (bounding boxes, halo-based movements, PDK-driven DRC checking, and netlist-parametric designs) that transfer cleanly to other analog systems such as ADC sub-blocks or sensor-interface modules. While exploring very large circuits is computationally more demanding, the pipeline itself is not limited by design principles but by available computing power. OSIRIS therefore provides a foundation upon which future work, both from our group and the community, can build to address broader classes of circuits and optimization tasks.

---

### Official Review · Reviewer_c6mf · 2025-10-31

**Soundness:** 2
**Presentation:** 2
**Contribution:** 3
**Rating:** 4
**Confidence:** 3

**Summary:**

This paper presents OSIRIS, a scalable dataset generation pipeline for analog IC layout design, producing over 64,200 circuit variations with detailed metrics to enable ML-driven research in EDA. It also introduces a reinforcement learning baseline that leverages the dataset for parasitic-aware layout optimization.

**Strengths:**

- The dataset is substantial, comprising more than 64,200 circuit variations, which could be highly valuable for future research.
- Since publicly available back-end analog circuit datasets are rare, this work has the potential to fill an important gap in the field.

**Weaknesses:**

- The main issue is that the paper is difficult to read. Given that ICLR is primarily an AI-focused venue, the paper should better explain the fundamental principles of analog back-end design and clearly describe the intended applications of the proposed benchmark. In its current form, it reads more like a technical report than a research paper.
- The experimental section is relatively short, even for a benchmark-oriented paper.

**Questions:**

- In Table 3, why does the Random method perform better than the open-source design tool MAGICAL?

---

> ### Author Response · Authors · 2025-11-24
>
> We thank the reviewer for the thoughtful comments and for recognizing the value of a large, parasitic-aware analog-layout dataset. We appreciate the opportunity to clarify the scope and contributions of OSIRIS and have revised the manuscript accordingly.
>
> We acknowledge that, for an AI-focused venue such as ICLR, the description of analog design principles should be more accessible. In the revised manuscript, we expanded the background discussion with a dedicated explanation of the analog front-end and back-end phases, highlighting why the back-end constitutes a bottleneck in the analog design process. This addition aims to bridge the gap between EDA-specific knowledge and the expectations of the ML community. Specifically, these aspects are discussed in Section 1 “Introduction”, lines 46/47 - 70/71.
>
> The framework serves a dual purpose: (i) enabling end-to-end parasitic-aware layout generation under different exploration strategies, including RL, and (ii) producing large volumes of DRC-free and LVS-verified annotated layouts for downstream ML tasks. To address the reviewer’s concern, we enriched the revised manuscript with a concrete dataset use case, showcasing how OSIRIS data can be employed to fine-tune a model for component-level layout generation. This demonstrates how the dataset facilitates practical ML workflows. The intended uses of OSIRIS dataset are detailed in new remarks at the end of Section 4 “Analog Layout Dataset”. While the details of the explored dataset use case are described in Section 6 “OSIRIS Dataset Use Case” and Appendix C “LLM Fine-tuning”.
>
> Regarding the brevity of the experimental section, we expanded it by incorporating the new dataset use case addressing component layouts generation. In particular, OSIRIS-generated samples are used to fine-tune an LLM to generate GDS layouts of capacitors. Section 6 “OSIRIS Dataset Use Case” and Appendix C “LLM Fine-tuning” provide in-depth details.
>
> Lastly, concerning the question about the performance of the Random method in Table 3, Section 7 “Experimental Results”, the discrepancy arises because OSIRIS employs its own internally designed placer and router. MAGICAL and ALIGN are not modular enough to reliably support iterative perturbations, repeated verification, and RL-driven refinement, which are central to OSIRIS. Our custom backend is optimized for consistency across thousands of variants and integrates DRC, LVS, and PEX tightly into the generation process. As a consequence, even a simple random-perturbation strategy operating on top of this infrastructure can, in some cases, produce layouts with lower pscore than single-pass outputs from external tools.

---

> > ### Comment · Reviewer_c6mf · 2025-11-28
> >
> > Thank you for the detailed clarifications and revisions. The expanded explanations of analog front-end and back-end design, the added dataset use case, and the strengthened experimental section have all improved the readability and accessibility of the paper for the AI community.
> >
> > Although the work does not employ the most recent AI techniques, the large-scale, well-structured, and parasitic-aware analog layout dataset presented here represents a meaningful and impactful contribution to the design automation community and is likely to benefit future research substantially.
> >
> > Given these improvements, I am raising my score by 2 points.

---

### Official Review · Reviewer_JYCi · 2025-11-01

**Soundness:** 2
**Presentation:** 3
**Contribution:** 2
**Rating:** 6
**Confidence:** 4

**Summary:**

The scarcity of open, high-quality datasets has constrained the use of machine learning in automating analog circuit design. This paper introduces OSIRIS, a scalable dataset-generation pipeline that uses reinforcement learning to systematically explore analog design spaces and produce DRC/LVS-clean layouts with comprehensive performance metrics and metadata, enabling robust benchmarking and generalizable ML methods.

**Strengths:**

1．	Introduces a dataset-generation pipeline for analog layouts and releases an open-source dataset augmented with post-layout simulations that guarantee the sample is LVS-, DRC-clean.

2．	Efficient design-space exploration. Proposes a reinforcement-learning-driven, iterative variant-generation method that enables efficient, performance-aware exploration of the analog layout space.

**Weaknesses:**

1．	Limited circuit type. The dataset currently covers only amplifier circuits at the 130 nm node.

2．	In Table 3, it’s not fair and confusing to compare with the MAGICAL and ALIGN, which are only analog layout generation tools without any design-space exploration.

3．	Constrained variant generation and diversity. Variants are created mainly by permuting device fingers and component placement within the halo, which limits structural diversity; some schematics permit fundamentally different layout topologies. Moreover, RL optimizes score and area only, without an explicit diversity objective, increasing the likelihood of many near-duplicate samples.

4．	Missing some usage examples. Include concrete examples of how the dataset can be used, e.g., training a post-layout performance predictor to guide place-and-route.

**Questions:**

1．	How do you handle cases where the RL agent fails to produce a valid solution after multiple iterations?

2．	Is component rotation included in the set of layout operations?

3．	Can the method scale to larger circuits like ADC/DAC?

---

> ### Author Response · Authors · 2025-11-24
>
> We thank the reviewer for the careful evaluation and constructive feedback. We appreciate the reviewer’s recognition of the importance of open, parasitic-aware analog layout datasets and the potential of OSIRIS to support machine-learning research in this domain.
>
> Regarding the limited circuit diversity, we agree that the current release focuses on four amplifier topologies implemented in the SkyWater 130 nm PDK. However, we note that the complexity of obtaining DRC- and LVS-clean layouts prevents the availability of open-source layout datasets, thus making our dataset extremely valuable to investigate novel ML-driven layout strategies. These circuits were intentionally selected because they are widely used in analog design and encompass heterogeneous device types, current-mirror structures, and differential pairs, thereby providing a representative and challenging layout space. Moreover, SkyWater 130 nm is a mature open-source PDK, fully supported by open-source CAD tools, which allows reproducibility and enables effective checking of DRC and LVS compliance. Nonetheless, OSIRIS was designed as a general framework, and we are actively extending it to additional circuit classes and technology nodes. To this end, Table 3, in Section 7 “Experimental Results”, has been enriched with a new circuit topology, namely low-pass filter (LPF), showcasing a different circuit family other than amplifiers.
>
> On the comparison with MAGICAL and ALIGN, our intention is not to claim that OSIRIS replaces these tools or that they share identical objectives. Rather, we aim to show that, when used as the solution space exploration strategy inside OSIRIS, reinforcement-learning-driven exploration can complement and outperform single-pass layout generation. Both MAGICAL and ALIGN represent state-of-the-art open-source layout frameworks; thus, evaluating OSIRIS-RL against their outputs provides a concrete and meaningful baseline. We added an extended clarification of this motivation to avoid any misinterpretation in initial remarks of Section 7 “Experimental Results” of the revised manuscript.
>
> Concerning variant diversity, we acknowledge the reviewer’s point and agree that our current exploration mechanism, finger permutations and spatial perturbations, does not cover the full structural variability allowed by some schematics. OSIRIS provides a first, extensible framework for generating DRC-clean and LVS-verified layouts and integrating learning-based refinement. Additional degrees of freedom, including rotation and more radical topological rearrangements, are planned extensions. Although each class of analog circuits relies on its own set of specialized quality metrics, the RL baseline methodology intentionally optimizes parasitics and area to keep the learning problem tractable and demonstrate feasibility. Nonetheless, the end-user can implement diversity-oriented objectives.
>
> On usage examples, we appreciate the suggestion. OSIRIS is intended not only to generate datasets but also to serve as an experimental platform for ML-driven analog layout research. Nevertheless, we enrich the revised manuscript with a concrete demonstration showing how OSIRIS-generated samples can be used to fine-tune an LLM to generate GDS layouts of components, in particular, focusing on capacitors generation. Details are described in Section 6 “OSIRIS Dataset Use Case” and Appendix C “LLM Fine-tuning”.
>
> Regarding the behavior of the RL agent when failing to produce valid layouts, OSIRIS integrates the agent with a deterministic place-and-route core that ensures robustness. If an RL-proposed perturbation results in a DRC, LVS, or simulation failure, the iteration terminates and the agents receive a penalty, while exploration continues from other trajectories. Because OSIRIS systematically explores the design space, valid solutions are quickly recovered.
>
> Component rotation is not yet included among the layout perturbation operations. We agree that it would expand the expressiveness of the search space and we plan to incorporate this degree of freedom in the next version of OSIRIS. We clarified the possibility to customize perturbation operations in the final remarks of Section 3.1 “Design Space Dimensions in Dataset Generation”.
>
> Finally, scalability to larger circuits such as ADCs or DACs is an important direction. We currently target compact analog blocks to ensure tractability in end-to-end parasitic extraction, verification, and RL-driven optimization. The pipeline, however, is not intrinsically limited to small circuits, and we are working toward supporting more complex and diverse designs.

---

### Author Response · Authors · 2025-11-24

In response to the reviewers' comments, we revised the manuscript to clarify the scope of OSIRIS, strengthen its technical positioning, and provide additional evidence of its practical utility for ML-driven analog design. The key points addressed in the rebuttal and the corresponding additions to the paper are summarized below:
1) Inclusion of an additional circuit and its exploration results, demonstrating the flexibility and extensibility of OSIRIS.
2) A use case was added to illustrate how the OSIRIS-generated data can be employed in practical ML-driven analog-design tasks.
3) The structure and clarity of several sections were improved to enhance readability and better highlight the manuscript’s main contributions.

Additions and edits are highlighted in blue in the revised manuscript.

---

### Meta-Review · Area_Chair_EdQo · 2026-01-04

**Summary:**

This paper introduces OSIRIS, a dataset generation pipeline for analog IC design, and releases a dataset of approximately 87,000 circuit variations (amplifiers at 130nm). It also proposes an RL-based baseline for analog design optimization. Reviewers generally agreed that the contribution fills a significant gap in the community regarding the scarcity of open, high-quality, post-layout analog circuit datasets (Reviewers c6mf, z7C5). The guarantee of LVS- and DRC-clean samples and the potential for scalability were also noted as strong points (Reviewer JYCi). The release of a large-scale, clean analog IC dataset is a commendable contribution that addresses a major bottleneck in ML for EDA. However, the limited scope of the circuits, questions regarding the fairness of baselines, and the need for writing improvement and clearer experimental validation of the dataset's utility weigh against the paper. While I recommend acceptance based on the resource value, I would not be opposed to rejection if the program constraints require higher standards of validation and scope.

**Reviewer Concerns:**

### **Partially Addressed Concerns**

* **Readability and Audience Fit (Partially Addressed):**
The authors explicitly mention improving the structure and clarity of sections to highlight main contributions. This directly responds to Reviewer c6mf’s concern about the paper reading like a technical report.
* **Missing Usage Examples & Downstream Validation (Partially Addressed):**
The inclusion of one specific use case demonstrates how the dataset can be used for ML-driven tasks. This addresses Reviewer JYCi’s request for "concrete examples" and Reviewer z7C5’s request for "verification in specific tasks."


* **Limited Scope and Generalizability (Partially Addressed):**
The addition of one extra circuit and its exploration results helps demonstrate the extensibility of the pipeline (answering "can it scale?"). However, this likely does not fully resolve the reviewers' fundamental concern that the *released dataset itself* is limited to 130nm amplifiers, nor does it necessarily prove generalizability to complex mixed-signal blocks like ADCs/DACs or advanced process nodes without more substantial data.

### **Unaddressed Concerns**

* **Efficiency and Diversity Limitations:**
The rebuttal do not mention any improvements to the generation speed (referenced as >50 hours per circuit by Reviewer z7C5) or the diversity of the layout topologies. The concern that the RL agent produces "near-duplicate samples" via simple permutations (Reviewer JYCi) appears to be unaddressed. This remains a significant validity risk for a dataset paper claiming to enable robust ML training.
* **Unfair Comparisons:**
There is no mention of modifying Table 3 or clarifying the comparison against MAGICAL and ALIGN. Reviewer JYCi’s specific critique that comparing a design-space exploration tool against pure layout generation tools is "unfair and confusing" has not been acknowledged or rectified.

**Reviewer Scores:**

According to the contents generated during the rebuttal process, I think Reviewer c6mf would have raised his/her score to 6. And the rest of reviewers would have maintained their original score.

---

### Decision · Program_Chairs · 2026-01-26

Accept (Poster)